# Dys-R Questionnaire: A Novel Screening Tool for Dysbiosis Linked to Impaired Gut Microbiota Richness

**DOI:** 10.3390/nu15194261

**Published:** 2023-10-05

**Authors:** Bianca Depieri Balmant, Danielle Cristina Fonseca, Ilanna Marques Rocha, Letícia Callado, Raquel Susana Matos de Miranda Torrinhas, Dan Linetzky Waitzberg

**Affiliations:** Laboratory of Nutrition and Metabolic Surgery of the Digestive System, LIM 35, Department of Gastroenterology, Hospital das Clínicas HCFMUSP, Faculdade de Medicina, Universidade de São Paulo, São Paulo 01246 903, Brazil; biancadepieribalmant@hotmail.com (B.D.B.);

**Keywords:** gastrointestinal microbiome, dysbiosis, chronic diseases, biodiversity, microbiota

## Abstract

Practical and affordable tools to screen intestinal dysbiosis are needed to support clinical decision making. Our study aimed to design a new subjective screening tool for the risk of intestinal dysbiosis from a previously described nonvalidated questionnaire (DYS/FQM) and based on subjective and objective data. A total of 219 individuals comprised the chronic diseases (CD; *n* = 167) and healthy control (HC; 52 subjects) groups. Sociodemographic, anthropometric, body composition, lifestyle, past history, intestinal health, and dietary data were collected. The gut microbiota (GM) profile was assessed from fecal samples using the 16S rRNA sequencing. Scores for the new tool (Dys-R Questionnaire) were assigned using discrete optimization techniques. The association between Dys-R scores and dysbiosis risk was assessed through correlation, simple linear models, sensitivity, specificity, as well as positive and negative predictive values. We found significant differences in the Chao1 Index between CD and HC groups (adjusted *p*-value = 0.029), highlighting lower GM richness as the primary marker for intestinal dysbiosis. DYS/FQM showed poor performance in identifying poor GM richness. Dys-R exhibited a 42% sensitivity, 82% specificity, 79% positive predictive value (PPV), and 55% negative predictive value (NPV) to identify poor GM richness. The new Dys-R questionnaire showed good performance in ruling out dysbiosis.

## 1. Introduction

In recent years, there have been significant advancements in understanding the composition of gut microbiota (GM) and its potential applications for human health [1,2].GM serves various vital roles in the host, including protection against pathogens [3], colonization of the mucosal surface [4], and production of a variety of antimicrobial substances [5]. These benefits are achieved through the homeostasis of bacteria-host relationship, a delicate balance that can be disrupted and lead to dysbiosis [6]. 

Although a precise definition of dysbiosis has not yet been established, the term is commonly used to describe an imbalance in the composition and function of the microbiota [7]. Dysbiosis can arise from overgrowth of pathobionts, depletion of commensal bacteria, or a decrease in microbial diversity, all of which can have negative effects on intestinal permeability, digestion, metabolism, and host immune responses [7]. Mounting clinical and experimental evidence supports the crucial role of dysbiosis-induced pro-inflammatory states in the development and progression of chronic diseases, including obesity [8], autoimmune diseases [9], inflammatory bowel diseases [10], and diabetes [11].

Various intrinsic and environmental factors can contribute to the development of dysbiosis by influencing the composition, diversity, and function of GM. These factors include diet [12], mode of birth [13], infant feeding method [14], medication usage (particularly antibiotics) [15], age [16], and the presence of underlying diseases [17]. Dietary interventions have demonstrated significant and rapid alterations in the GM composition [12,18]. Misuse of antibiotics disrupts beneficial microbes, leading to depletion of GM or overgrowth of undesirable bacteria [15,19]. Moreover, mental stress has been associated with physiological changes in the gastrointestinal tract, activating the hypothalamic–pituitary–adrenal axis, compromising mucosal immune functions, and resulting in dysbiosis of GM [20].

Interventions targeting GM hold promise for modifying community composition and preventing and treating diseases associated with dysbiosis [21]. GM evaluation methods, such as metagenomic analysis, can screen for dysbiosis [22], but their clinical application is limited due to complexity and cost [22,23]. There is a growing need for practical, affordable tools to identify dysbiosis risk, aiding clinical decision making and targeted therapeutic interventions for microbiota modulation.

One promising avenue in this field is the development of subjective screening tools that can reflect the risk of intestinal dysbiosis. Such tools would enable the identification of individuals who would benefit from further detailed analysis of the GM’s genetic profile while also assisting in the evaluation of modifiable risk factors associated with dysbiosis. To the best of our knowledge, the only questionnaire that aims to fulfill this purpose is the Dysbiosis Frequent Questions Management (DYS/FQM; Appendix A), registered at the National Institute of Industrial Property (INPI-914742353) by Laboratório Farmoquímica^®^. However, this tool lacks scientific validation in the literature, specifically in terms of directly comparing its performance to the genetic sequencing of microbiota species present in fecal samples of individuals.

The DYS/FQM was developed based on risk factors associated with dysbiosis reported in the literature [12,13,14,15,16,17,18,19,20], encompassing lifestyle, medical history, gut health, and dietary habits. Each variable in the questionnaire was assigned an arbitrary score based on its presence, absence, or frequency of exposure. By utilizing objective data obtained from intestinal metagenomics as a reference, our study aimed to assess and improve the performance of the DYS/FQM questionnaire. The ultimate objective was to develop a new and more effective subjective screening tool capable of accurately identifying the risk of intestinal dysbiosis. 

## 2. Materials and Methods

### 2.1. Study Design and Subjects

This study is a prospective, observational, and cross-sectional single-institution investigation approved by the local ethics committee (CaPPesq 3.008.966). Before the commencement of this study, written informed consent was obtained from all participants, adhering to the guidelines outlined in the Declaration of Helsinki.

A total of 167 patients with chronic diseases (CD) and 52 healthy volunteers were recruited for this study. The inclusion criteria for DC patients were as follows: age over 18 years old, no recent use of acute medication within one month before fecal sample collection, and regular outpatient follow-up at the Hospital das Clínicas, Faculdade de Medicina da Universidade de São Paulo (HC-FMUSP) for the treatment of one of the following chronic diseases: obesity (body mass index > 30 kg/m^2^; *n* = 24), type 2 diabetes (*n* = 20), type 1 diabetes (*n* = 20), Crohn’s disease (*n* = 20), ulcerative colitis (*n* = 20), systemic lupus erythematosus (*n* = 23), rheumatoid arthritis (*n* = 20), and psoriasis (*n* = 20). The diagnosis of all patients was confirmed through appropriate methods such as laboratory tests, imaging studies, endoscopic procedures, and/or histopathological examinations. Participants who met any of the following criteria were excluded from this study: viral diseases, pregnancy or lactation, antibiotic use in the last month, ingestion of prebiotics or probiotics within the past two months, use of laxatives, abdominal surgery within the past six months, and cognitive deficits that would hinder their participation in this study. 

The inclusion criteria for the Healthy Control (HC) group were as follows: individuals aged over 18 years with no history of chronic medication use (except for contraceptives), absence of acute medication use within one month before the fecal sample collection, and the absence of any previously diagnosed diseases. Healthy participants who met any of the following items were excluded from this study: pregnancy or lactation, intake of prebiotics or probiotics in the last 2 months, use of laxatives, special dietary habits or food restrictions, drug or alcohol abuse, history of radiotherapy/chemotherapy treatment, previous gastrointestinal surgery (excluding appendectomy), body mass index (BMI) greater than 30 kg/m^2^, and cognitive deficits that would compromise their participation in this study. Upon enrollment, all selected participants underwent the collection of sociodemographic and anthropometric data, body composition assessment, and investigation of lifestyle factors, past medical history, intestinal health, and dietary habits. Additionally, fecal samples were collected from each participant for the analysis of GM.

### 2.2. Participants’ Characterizing Data

For both groups, a trained researcher administered a semistructured questionnaire to collect sociodemographic data. The anthropometric assessment included measurements of weight and height to calculate the BMI, which was determined by dividing each individual’s weight by the square of their height and expressed in kg/m^2^. Considering its relationship with GM, data on body composition were also evaluated using a portable electric bioimpedance device (Quadiscam 4000, Bodystat Ltda^®^, Douglas, Isle of man, British Isles) operating at four frequencies (5, 50, 100, 200 kHz).

### 2.3. Evaluation of the Gut Microbiota (GM)

Fecal samples were obtained in 2 mL microtubes containing a DNA preservation buffer solution by participant self-collection using a sterile swab. GM sequencing was performed by obtaining fecal DNA and amplifying the V3 and V4 region of the 16S rRNA gene, as detailed in International Human Microbiome Standards (IHMS) SOP06 (http://www.microbiome-standards.org; 2 July 2021) using the MiSeq platform (Illumina Miseq^®^, San Diego, CA, USA). The bioinformatics analysis of the 16S rRNA data was performed at the Bioinformatics Platform of the Rene Rachou Institute, Fiocruz Minas (Belo Horizonte, MG, Brazil). After removing sequences with more than 2 expected errors, each ASV had its taxonomic classification assigned by the TAG.ME package [24] using the specific model to correspond to the V3 and V4 regions. Taxonomic identifications of bacteria were assigned to each ASV using DADA2 [25], based on exact correspondence between ASVs and the reference sequences in the Silva database (version 132) [26].

### 2.4. Development of the Dysbiosis Risk Questionnaire (Dys-R Questionnaire)

Participants were asked to complete the DYS/FQM questionnaire (Appendix A) aiming to 1. assess for the first time its performance in identifying dysbiosis based on objective GM measurements (validation) and 2. support the development of the Dysbiosis Risk Questionnaire (Dys-R) based on its findings. Therefore, after the validation of the DYS/FQM tool, questions were adapted into binary answers (yes or no), with a “yes” answer indicating exposure to a known harmful factor to GM. The participants’ answers were reassigned as follows: being born by cesarean section, breastfeeding for less than 6 months, consuming less than 5 daily servings of fruits, vegetables, legumes, and/or whole grains, consuming refined sugar or artificial sweeteners more than once a day, consuming ultraprocessed foods more than 3 times a week, consuming more than 4 doses of alcoholic beverages per week, engaging in less than 150 min of physical activity per week, experiencing high psychological stress, smoking, using antibiotics or nonsteroidal anti-inflammatory drugs in the past 3 months, taking more than 3 medications continuously, undergoing treatment or monitoring for any health condition, having three or more liquid bowel movements per day or experiencing difficult bowel movements (such as hard stools and/or less than 3 bowel movements per week), and having undergone a medium or major surgical procedure in the last 60 days or bariatric surgery at any time in life. 

These questions with binary answers were used to construct the Dys-R model. The differences in the composition of GM were considered as an outcome in the Dys-R questionnaire. The performance of the Dys-R questionnaire in identifying dysbiosis based on objective GM measurements was also assessed. 

### 2.5. Statistical Analysis

Continuous variables are presented as mean and standard deviation, and categorical variables are presented as absolute and relative frequencies. The normality of continuous variables was assessed using the Shapiro–Wilk test. Comparisons between groups of the anthropometric and body composition data were performed using the Student T-test or Mann–Whitney tests, and comparisons between groups of the lifestyle factors, past medical history, intestinal health, and dietary habits were performed using the Chi-square test.

Simpson, Shannon, and Chao1 indices for alpha (α) diversity were calculated using the vegan R package (version 2.5-7), while population beta (β) diversity was calculated by Jensen–Shannon divergence. The PERMANOVA (Permutation Multivariate Analysis of Variance) analysis with *p*-value adjusted by Benjamini–Hochberg (FDR = 0.05) was also performed to identify the significant difference between groups in the macrostructure of the bacterial population.

In the development of the Dys-R Questionnaire, the cutoff point for the dysbiosis risk variable was determined based on quartiles obtained from the CG group. The sensitivity and specificity of the capacity for identification of dysbiosis risk were calculated using data from 219 individuals, considering the binary variables “Healthy/Diseased” and “High/Low Richness”. To assign scores to each question, discrete optimization techniques were applied so that a score between 0 and 4 was given to “yes” responses, whereas “no” responses received a score of 0.

To assess the correlation between the Dys-R or DYS/FQM score and the dysbiosis risk variable, an analysis using the Pearson test was performed. Simple linear models were constructed, and sensitivity, specificity, as well as positive and negative predictive values were calculated. The accuracy of the Dys-R was evaluated by analyzing the Receiver Operating Characteristic (ROC) curve and calculating the area under the curve (AUC).

For all analyses, specific packages of the R software (version 4.2) and Jasp (version 0.17.1) were used, a significance level of 5% (*p* < 0.05), and a 95% confidence interval (IC) was adopted.

## 3. Results

### 3.1. Patient’s Descriptive Data 

This study included a total of 219 individuals with a mean age of 45.41 ± 15.17 years. As presented in Table 1, there were variations in sociodemographic, anthropometric, and body composition characteristics among the groups.

The CD group demonstrated a higher prevalence of several lifestyle factors, past medical history, intestinal health, and dietary habits compared to the HC group (Table 2). Specifically, individuals in the CD group had a higher prevalence of being born by cesarean section birth, consuming refined sugar or artificial sweeteners more than once a day, engaging in physical activity for less than 150 min per week, recent antibiotic use in the past 3 months, consuming more than 3 medications continuously, undergoing treatment or monitoring for health conditions, and experiencing changes in stool patterns (liquid and hard).

### 3.2. Gut Microbiota (GM)

The Chao1 Index showed a significant difference between the CD and HC groups (adjusted *p*-value = 0.029). However, no statistically significant differences were observed between the groups in the other parameters of alpha diversity (Figure 1).

Regarding the beta diversity analysis, although a 16.9% difference in the overall structure of the bacterial population was observed between the groups (Figure 2), this dissimilarity did not reach statistical significance (R^2^ = 0.006; adjusted *p*-value = 0.07), and the betadisper results were not significant (*p* = 0.336). Additionally, the Firmicutes/Bacterioidetes (F/B) ratio in the CD group did not show a significant difference compared to the HC group (0.932 ± 0.805 vs. 0.899 ± 0.651; *p* < 0.985, Mann–Whitney test).

The GM phyla most predominant were Bacteroidetes, Firmicutes, Proteobacteria, Verrucomicrobia, Actinobacteria, Fusobacteria, Lentisphaerae, Synergistetes, Elusimicrobia, Tenericutes, and Epsilonbacteraeota (Figure 3). Notably, no significant differences were observed between the groups concerning these phyla.

### 3.3. DYS/FQM Performance

Considering the significant and unique relevance of the Chao1 Index to segregate CD from HC, this measure was selected as the primary outcome measure to identify the risk of dysbiosis. The cutoff point for the Chao1 Index was determined based on quartiles obtained from the HC group. Consequently, the quartile that demonstrated the highest ability to identify individuals with the condition of interest was the 25th quartile (Chao1 Index = 203.65; Sensitivity = 0.8077; Specificity = 0.2695; *p* < 0.001). The score obtained in the DYS/FQM questionnaire by the CD group was significantly different from that of the HC group (*p* = 0.006) and showed a weak inverse correlation with the Chao1 index (*r* = −0.1838; CI = −0.3089, −0.0525). The sensitivity and specificity of the DYS/FQM questionnaire were 55% and 55%, respectively, with a PPV of 29% and an NPV of 88% (AUC = 0.61; 95% CI = 0.52–0.7; *p* < 0.001; Appendix A).

### 3.4. Dys-R Construction

We developed the Dys-R model using binary response questions derived from the adapted DYS/FQM questionnaire focusing on lifestyle factors, past medical history, intestinal health, and dietary habits. To assign scores to each question, discrete optimization techniques were applied. A score between 0 and 4 was given to “yes” responses, while “no” responses received a score of 0. Questions that received a score of 0 for a “yes” response were eliminated from the questionnaire as they did not contribute to assessing the richness of individuals’ GM. The included eliminated questions: consuming less than five daily servings of fruits, vegetables, legumes, and/or whole grains; consuming refined sugar or artificial sweeteners more than once a day; consuming more than four alcoholic beverages per week; practicing less than 150 min of physical activity per week; having a high perception of psychological stress; taking more than three medications continuously; having three or more liquid bowel movements per day or experiencing difficult bowel movements (such as hard stools and/or less than three bowel movements per week); and undergoing a medium or major surgical procedure in the last 60 days or bariatric surgery at any time in life. As a result, the questionnaire was refined, resulting in a final set of seven questions.

### 3.5. Dys-R Validation 

The questionnaire, composed of seven questions with scores assigned by discrete optimization (Figure 4), was evaluated for its performance in identifying the low richness of GM by analyzing the receiver operating characteristic (ROC) curve using the 25% quartile of the Index Chao1 (Figure 5). The sensitivity and specificity of this questionnaire were determined to be 42% and 82%, respectively, with a positive predictive value (PPV) of 79% and a negative predictive value (NPV) of 55%. The identified score cutoff point was 8, indicating that individuals with a final score above 8 in the questionnaire would be at risk of dysbiosis, characterized by low GM richness (AUC = 0.65; 95% CI = 0.56–0.73; *p* < 0.001). Additionally, there was a significant association between the Dys-R score and the Chao 1 Index (*r* = −0.3206; 95% CI = (−0.4347, −0.1964); *p* < 0.0001).

## 4. Discussion

In the present study, our findings revealed that individuals with chronic diseases exhibited lower GM richness, as assessed by the Chao1 index, compared to healthy controls. However, we found no difference when examining other parameters of GM composition and bacterial taxon abundance at the phylum level between the two groups. These results highlight the complexity surrounding fluctuations in the GM composition, as even in distinct health conditions, the macrostructure of GM was not significantly different between the groups. Furthermore, our findings suggest that GM composition is influenced by multiple factors, including intrinsic characteristics of the individual, as well as the combined effects of environmental, social, dietary, lifestyle factors, and others. The complex interplay of these factors likely contributes to the unique individualized nature of GM composition, making it challenging to identify a definitive healthy or dysbiotic pattern [27,28].

Several studies have explored specific GM characteristics in relation to health markers, uncovering correlations between specific phyla and genera of intestinal microbes. Nevertheless, the fundamental characteristics of a bacterial community lie in its numerical composition and diversity [29,30], as biodiversity plays a crucial role in maintaining a balanced ecosystem [31]. In fact, investigations of GM patterns report that the deterioration of human health is often associated with reduced measures of alpha diversity, such as the Chao1 index, which aligns with the findings of the present study [31,32]. Therefore, GM richness may be a more significant indicator of intestinal health than phyla and genera due to its collective function, functional redundancy, and ecosystem stability.

It is important to emphasize that the presence of dysbiosis, as measured in this study by the Chao1 index, does not necessarily imply a causal relationship with the evaluated diseases. Instead, there is evidence suggesting that changes in GM often result from a combination of various factors, including diet [12], medication use [33], physical activity [34], age [16], and others. Although the independent impacts of these factors on GM was not specifically evaluated in our study, it is plausible to propose that a detrimental combination of these factors may contribute to the reduction in GM richness. This hypothesis is further supported by the results obtained in our study through the construction of the Dys-R questionnaire.

Given the dynamic nature of research in the GM area, ongoing efforts are being made to develop new approaches and methods that enhance our understanding and clinical application of dysbiosis. Currently, various dysbiosis indices have been proposed to assist professionals in interpreting the GM health status. However, these indices differ significantly in their methodologies, cohort selection, and targeted conditions. It generally requires genetic sequencing of the microbiota composition and utilizes diverse approaches such as large-scale bacterial marker profiles, taxa-based methods, nearest-neighbor classification, random forest prediction, and combined analysis of alpha–beta diversity. While these indices contribute to our knowledge, their clinical applicability is limited [27]. In addition to these indexes, there is mention in the literature of the ongoing development of a multiplatform Health app (mHealth) for dysbiosis detection (mDys), which takes a multiomic approach, specifically for individuals with inflammatory bowel disease (IBD) [35]. Other fecal-based tests and algorithms have also been created, primarily focused on IBD, and typically require the collection of fecal samples [36].

Considering the limitations and the need for practical and accessible tools, our study proposes a screening tool with high specificity and positive predictive value for identifying low GM richness in individuals with different health conditions. We emphasize the importance of assessing risk factors for dysbiosis and the necessity of controlling modifiable factors, given that an imbalance of GM can have detrimental consequences, such as disruption of the intestinal barrier and compromise of the host’s immune and metabolic systems [37].

To date, there are a limited number of studies investigating dysbiosis screening questionnaires designed for clinical practice at low cost. Furthermore, the available questionnaires have significant limitations that need to be considered before their implementation. For instance, Okipney et al. [38] utilized the National Dysbiosis Survey (INDIS) instrument to evaluate the association between epidemiological and clinical factors and the risk of dysbiosis in hospitalized adult patients. However, this questionnaire, which employs arbitrary scores for each variable, has not yet been validated through direct comparison with genetic sequencing of species found in fecal samples from individuals with dysbiosis-related variables. Similarly, Rodrigues et al. [39] performed an exploratory theoretical analysis of the dysbiosis risk in individuals with IBD using the DYS/FQM questionnaire. It is worth mentioning that in our results, although this questionnaire showed statistical significance, the inverse correlation was weak and had low positive predictive power in identifying dysbiosis associated with GM richness.

The current scenario underscores the uniqueness of the Dys-R questionnaire in the clinical context for screening dysbiosis associated with low GM richness. It can serve as a valuable tool in supporting clinical decision making and facilitating the implementation of therapeutic approaches targeted at GM modulation. Although our study has limitations, such as reliance on participant-provided responses that may be subject to memory bias, interpretation bias, or subjectivity, and the inclusion of questions assessing subjective aspects like symptoms and stress level, the entire team of researchers was equally trained to standardize procedures and rigorous criteria were applied for data collection. By providing a practical and efficient means of identifying individuals at risk of dysbiosis associated with low GM richness, the Dys-R questionnaire holds promise to be a valuable tool in clinical practice. Further studies are encouraged to evaluate the performance of the Dys-R questionnaire in the context of overall health and specific population groups, aiming to validate and expand upon our findings.

## 5. Conclusions

The Dys-R questionnaire, based on a refined set of questions, presents a novel and promising approach for dysbiosis screening associated with low GM richness in a clinical setting. It stands out for its high specificity and positive predictive value, which are essential attributes in a screening tool. The user-friendly nature of the Dys-R questionnaire supports its applicability.

## Figures and Tables

**Figure 1 nutrients-15-04261-f001:**
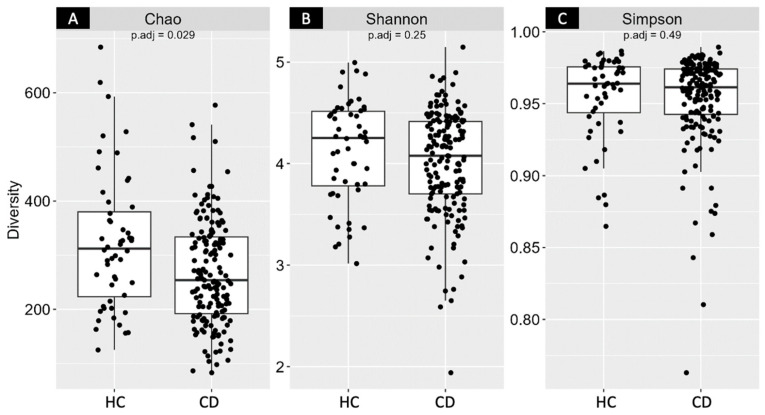
Comparison of bacterial alpha diversity represented by (**A**) Chao1 Index, (**B**) Shannon Index, and (**C**) Simpson Index and of the Healthy Control group (HC; *n* = 52) versus Chronic Disease group (CD; *n* = 167). Statistical significance adjusted *p*-value < 0.05.

**Figure 2 nutrients-15-04261-f002:**
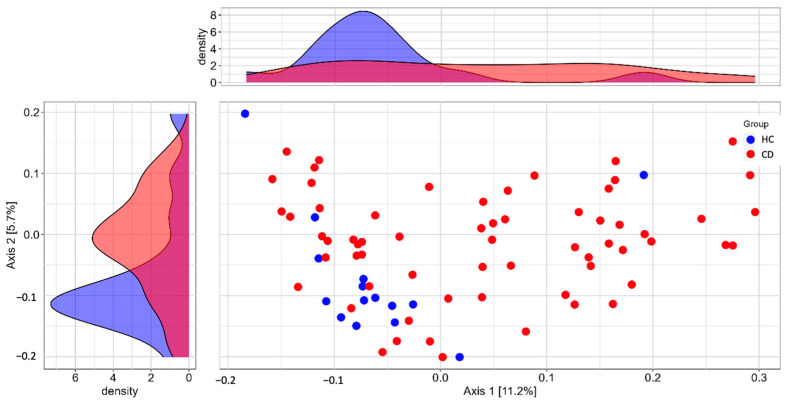
Principal coordinate analysis (PCoA) of the gut microbiota macrostructure based on the Jensen–Shannon index, comparing Healthy Control group (HC; *n* = 52) versus Chronic Disease group (CD; *n* = 167).

**Figure 3 nutrients-15-04261-f003:**
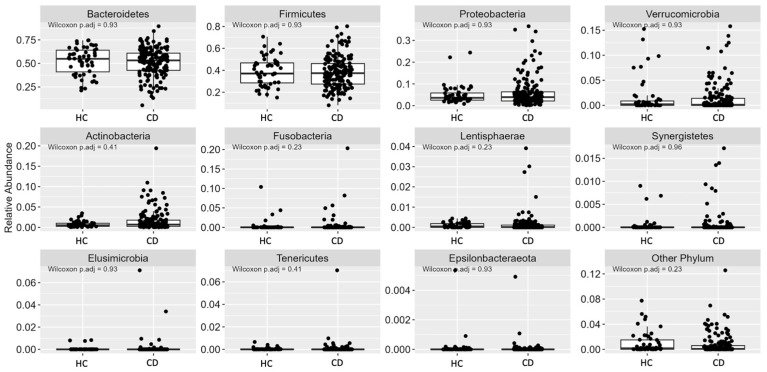
Relative abundance (%) of intestinal bacterial phyla in Healthy Control (HC; *n* = 52) versus Chronic Disease group (CD; *n* = 167). Statistical significance *p*-adjusted value < 0.05.

**Figure 4 nutrients-15-04261-f004:**
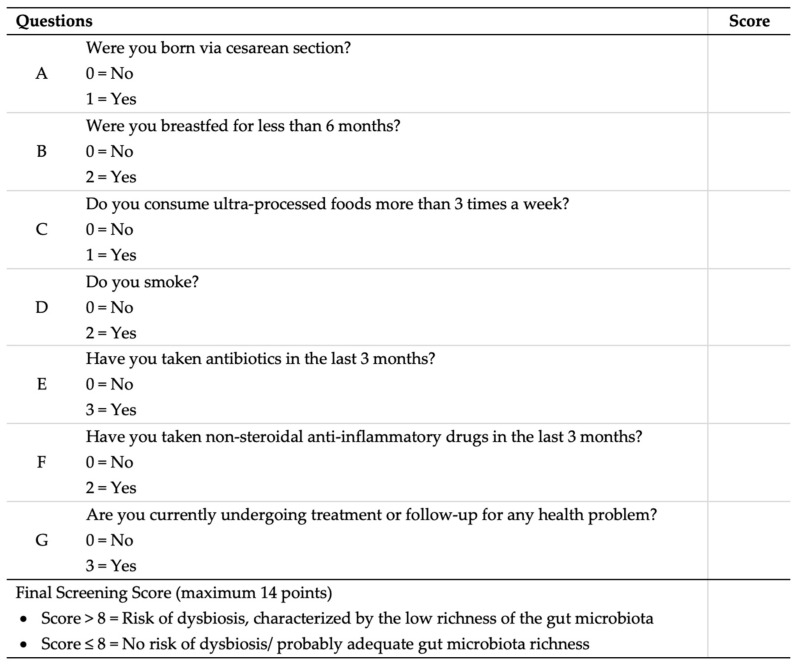
Dysbiosis Risk Questionnaire (Dys-R Questionnaire) with values attributed by discrete optimization.

**Figure 5 nutrients-15-04261-f005:**
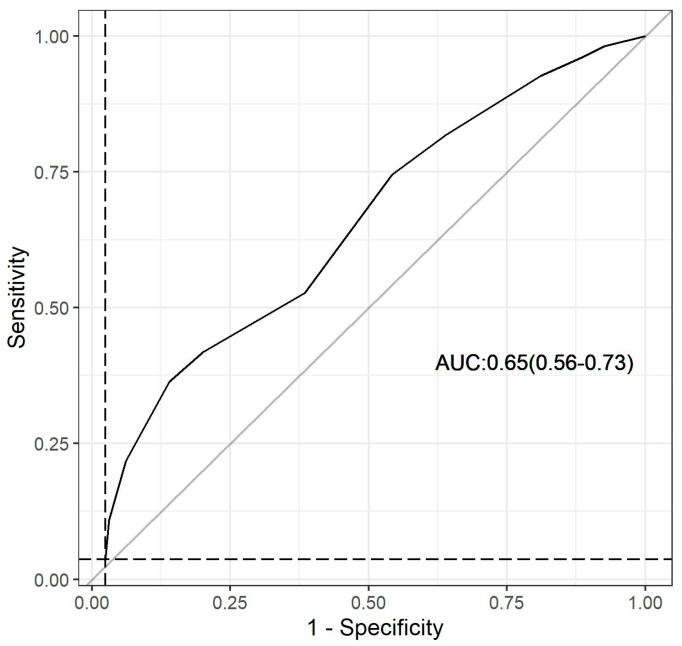
Receiver Operating Characteristic (ROC) curve of the Dysbiosis Risk Questionnaire (Dys-R Questionnaire), considering the quartile 25% of the Chao1 Index.

**Table 1 nutrients-15-04261-t001:** Characterization of participants according to sociodemographic, anthropometric, and body composition variables.

Variables	CD ^1^ (*n* = 167)	HC ^2^ (*n* = 52)	*p*-Value
Age, years	47.0 ± 13.9	40.4 ± 17.9	<0.001 ^3^
Gender			
Female, *n* (%)	111 (66.5)	23 (44.2)	0.004 ^4^
Male, *n* (%)	56 (33.5)	29 (55.8)	
Ethnicity			
Black, *n* (%)	21(12.6)	3 (5.8)	<0.001 ^4^
Brown, *n* (%)	61 (36.5)	4 (7.7)	
White, *n* (%)	84 (50.3)	39 (75.0)	
Yellow, *n* (%)	0 (0)	6 (11.5)	
Indigenous, *n* (%)	1 (0.6)	0 (0)	
Anthropometric			
Weight, kg	79.8 ± 26.5	68.6 ± 13.1	0.014 ^3^
Height, cm	164 ± 0.1	167 ± 0.1	0.001 ^3^
BMI ^5^, kg/m^2^	29.6 ± 9.2	23.9 ± 2.9	<0.001 ^3^
Body composition			
LMP ^6^, %	61.4 ± 11.3	72.4 ± 7.5	<0.001 ^3^
FMP ^7^, %	38.6 ± 11.3	27.6 ± 7.5	<0.001 ^3^

^1^ CD = Chronic Diseases. ^2^ HC = Healthy Control. ^3^
*t*-test or Mann–Whitney test. ^4^ Chi-square test. ^5^ BMI = Body Mass Index. ^6^ LMP = Lean Mass Percentage. ^7^ FMP = Fat Mass Percentage. Statistical significance *p* < 0.05.

**Table 2 nutrients-15-04261-t002:** Characterization of participants according to lifestyle factors, past history, intestinal health, and dietary habits.

Variables, *n* (%)	CD ^1^ (*n* = 167)	HC ^2^ (*n* = 52)	*p*-Value ^3^
Age ≥ 60 years	39 (23.4)	9 (17.3)	0.358
Born by cesarean section	30 (18.0)	25 (48.1)	<0.001
Breastfed for less than 6 months	104 (62.3)	32 (61.5)	0.924
Consumption of less than 5 daily servings of fruits, vegetables, legumes, and/or whole grains	155 (92.8)	44 (84.6)	0.073
Consumption of refined sugar or artificial sweeteners more than once a day	156 (93.4)	40 (77.0)	<0.001
Consumption of ultraprocessed foods more than 3 times a week	75 (44.9)	19 (36.5)	0.287
Consumption of greater than 4 doses of alcoholic beverage per week	2 (1.2)	3 (5.8)	0.054
Less than 150 min of physical activity per week	135 (80.8)	30 (57.7)	<0.001
High self-reported psychological stress	56 (33.5)	11 (21.2)	0.091
Smoking	16 (9.6)	1 (1.9)	0.072
Use of antibiotics in the last 3 months	45 (27.0)	5 (9.6)	0.009
Use of nonsteroidal anti-inflammatory drugs in the last 3 months	38 (22.8)	7 (13.5)	0.148
Continuous consumption of more than 3 medications	74 (44.3)	0 (0)	<0.001
Current treatment or monitoring for any health condition	167 (100)	52 (100)	<0.001
Three or more liquid bowel movements per day or difficult bowel movements, hard stools, and/or less than 3 bowel movements per week	44 (26.4)	5 (9.6)	0.011
Medium or major surgical procedure in the last 60 days or bariatric surgery at any time in life	2 (1.2)	1 (1.9)	0.694
Treatment with chemotherapy or radiotherapy			NA ^4^

^1^ CD = Chronic Diseases. ^2^ HC = Healthy Control. ^3^ Chi-square test. Statistical significance *p* < 0.05. ^4^ NA = not applicable. Statistical significance *p* < 0.05.

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
