# Peer review of "Dys-R Questionnaire: A Novel Screening Tool for Dysbiosis Linked to Impaired Gut Microbiota Richness"

_nutrients, 2023, doi:10.3390/nu15194261_

Round 1
Reviewer 1 Report (Previous Reviewer 2)
I am satisfied with the revised manuscript. No further comments.
Author Response
Thank you for taking the time to review the manuscript and for your kind feedback. We are delighted to hear that you are satisfied. We greatly appreciate your dedication to the peer-review process, which plays a crucial role in advancing scientific knowledge.
Reviewer 2 Report (Previous Reviewer 1)
Authors reported a new questionnaire, Days-R, to screen risk for dysbiosis, which is meaningful. It makes sense that bad daily habits increase the risk of dysbiosis and chronic diseases. But the correlation between lifestyle, past history, and Intestinal and dietary habits with gut microbiota is kind of weak, since there are not much significant differences of GM phyla between chronic and healthy groups (Figure 3).
And authors collect data from several kinds of chronic diseases patients, such as obesity, Crohn's diseases, psoriasis and so on, it's not very clear which is the main contribution to dysbiosis, could authors add analysis of gut microbiota of each group of patients, this would be more indicative.
Author Response
Please see the attachment.

This manuscript is a resubmission of an earlier submission. The following is a list of the peer review reports and author responses from that submission.
Round 1
Reviewer 1 Report
The authors established the correlation between dysbiosis and Dys-R Questionnaire, which would provide improvement to clinical diagnosis.
And there are some concerns:
1. there is no significant differences between CD and HC groups in microbiota diversity according to Shannon Index and Simpson Index (Figure 1); and no significant differences in main phyla (Figure 3). So, the causation of microbiota and dysbiosis is not strong enough. The patients' descriptive data and Dys-R questionnaire showed HC group have better dietary habits, which makes sense, but do these factors contribute to microbiota diversity and richness? This part needs more clarification and explanation.
2. The questionnaire composed of 7 simple questions (Figure), it is not very delicate in my opinion.
The English quality is good!
Author Response
1. There is no significant differences between CD and HC groups in microbiota diversity according to Shannon Index and Simpson Index (Figure 1); and no significant differences in main phyla (Figure 3). So, the causation of microbiota and dysbiosis is not strong enough. The patients' descriptive data and Dys-R questionnaire showed HC group have better dietary habits, which makes sense, but do these factors contribute to microbiota diversity and richness? This part needs more clarification and explanation.
Thank you for reviewing our work and for your considerations regarding the presented results. Below, we provide a detailed response to address your questions and concerns:
- We acknowledge your observation that no significant differences were found between the CD and HC groups regarding microbiota diversity, as indicated by the Shannon and Simpson indices. However, it is worth noting that the Chao1 Index showed a significant difference between the CD and HC groups (adjusted p-value = 0.029). The Chao, Shannon, and Simpson indices are all metrics used to assess different aspects of microbial community diversity in the human gut. Each index approaches diversity in distinct ways and provides valuable information about the composition and species richness. While the Shannon and Simpson indices take into account evenness (the relative distribution of species), the Chao1 index considers rare species. Thus, the calculation of the Chao1 Index is based on counting unique species and the abundance of species occurring in more samples but not in all samples. Since in our study, the difference in the overall structure of the bacterial population between the groups is small (16.9%; beta diversity results), the Chao1 metric becomes a valuable tool for characterizing the gut microbiota and can significantly contribute to interpreting results in studies involving microbial diversity analysis. We hope this detailed explanation provides a clearer understanding of our methodology and justifies the inclusion of the Chao1 Index as a relevant measure in our study.
- We acknowledge that the HC group (healthy habits group) demonstrates better dietary habits, which may influence, along with other factors, the richness of the gut microbiota. During the discrete optimization analysis to determine the weight of each variable on the gut microbiota, questions that received a score of 0 for a "yes" response were excluded from the questionnaire as they did not contribute to assessing the microbiota richness of the individuals. The eliminated questions included: consumption of less than 5 daily servings of fruits, vegetables, legumes, and/or whole grains; daily intake of refined sugar or artificial sweeteners; consumption of more than 4 alcoholic beverages per week, among others. From this analysis, we also identified that consuming ultra-processed foods more than 3 times per week, in conjunction with other lifestyle factors, significantly impacted the Chao richness of the gut microbiota. Therefore, this question was included in the Dys-R questionnaire.
2. The questionnaire composed of 7 simple questions (Figure), it is not very delicate in my opinion.
We agree that the identification of intestinal dysbiosis is a much more complex process than simply using 7 simple questions. However, it is important to emphasize that the objective of the Dys-R questionnaire is not to be a diagnostic tool. Instead, we highlight the uniqueness of the Dys-R questionnaire in the clinical context, where it can be used as a screening tool to identify individuals susceptible to dysbiosis associated with low richness of the gut microbiota. In this way, we believe that the Dys-R questionnaire can be a valuable tool to support clinical decision-making, facilitating the implementation of therapeutic approaches targeted at modulating the gut microbiota. For example, it can be used to filter individuals who need a more in-depth investigation of their gut microbiota composition, considering that diagnostic methods can be expensive.
Reviewer 2 Report
The study should be of interest for the readers of Nutrients and the data are well presented.
I lack some minor details that should be added.
Line 94: obesity. Which definition was used? BMI>30? Please add
Line 94-115. There are specific criteria for inclusion regarding prebiotics and pro-biotics but no comment on antibiotics. Most likely there were criteria for antibiotics use prior to inclusion. No recent use of acute medication could possibly cover antibiotics use, but it does not rule out long-time antibiotic use. Considering the potent effect of antibiotics on the GM I think it should be more clearly specified.
Line 172: Benjamini-Hochberg can be used with different false discovery rates. Please add the FDR. 0.04? 0.1?
A personal reflection is that the numbers is very precise with often 4 figures. Is for instance really the accuracy of the weight that good that you can measure it at 10 g accuracy (79.82 kg)? I would have preferred 79.8 kg, but that is a personal opinion. It is more of a feedback and I will accept if you want to keep the present layout as long as the editor think it is OK.
Author Response
1. Line 94: obesity. Which definition was used? BMI > 30? Please add
Thank you for providing your valuable comment and suggestion. We want to assure you that we carefully considered your recommendation and have made the necessary improvement by including the definition of BMI > 30 kg/m2 in the manuscript.
2. Line 94-115. There are specific criteria for inclusion regarding prebiotics and pro-biotics but no comment on antibiotics. Most likely there were criteria for antibiotics use prior to inclusion. No recent use of acute medication could possibly cover antibiotics use, but it does not rule out long-time antibiotic use. Considering the potent effect of antibiotics on the GM I think it should be more clearly specified.
Thank you for your valuable comment. We excluded participants who had used antibiotics within one month before fecal sample collection. Therefore, even if a patient had a chronic or acute antibiotic use, if this consumption occurred within one month before fecal sample collection, they were not included in the study. We included this information in the exclusion criteria.
3. Line 172: Benjamini-Hochberg can be used with different false discovery rates. Please add the FDR. 0.04? 0.1?
Thank you for your valuable suggestion. We would like to assure you that we have taken your recommendation into consideration and have included the FDR value (0.05).
4. A personal reflection is that the numbers is very precise with often 4 figures. Is for instance really the accuracy of the weight that good that you can measure it at 10 g accuracy (79.82 kg)? I would have preferred 79.8 kg, but that is a personal opinion. It is more of a feedback and I will accept if you want to keep the present layout as long as the editor think it is OK.
Thank you for your valuable suggestion. We would like to assure you that we have taken your recommendation into consideration and have made the formatting changes throughout the entire text.